# The Controversial Role of HCY and Vitamin B Deficiency in Cardiovascular Diseases

**DOI:** 10.3390/nu14071412

**Published:** 2022-03-28

**Authors:** Wolfgang Herrmann, Markus Herrmann

**Affiliations:** 1Medical School, Saarland University, 66421 Saarbrücken, Germany; w-herrmann@gmx.de; 2Clinical Institute of Medical and Chemical Laboratory Diagnostics, Medical University of Graz, 8036 Graz, Austria

**Keywords:** homocysteine, B vitamins, B-vitamin supplementation, telomere, telomere shortening, vascular dysfunction, atherosclerosis, cardiovascular disease

## Abstract

Plasma homocysteine (HCY) is an established risk factor for cardiovascular disease CVD and stroke. However, more than two decades of intensive research activities has failed to demonstrate that Hcy lowering through B-vitamin supplementation results in a reduction in CVD risk. Therefore, doubts about a causal involvement of hyperhomocysteinemia (HHcy) and B-vitamin deficiencies in atherosclerosis persist. Existing evidence indicates that HHcy increases oxidative stress, causes endoplasmatic reticulum (ER) stress, alters DNA methylation and, thus, modulates the expression of numerous pathogenic and protective genes. Moreover, Hcy can bind directly to proteins, which can change protein function and impact the intracellular redox state. As most mechanistic evidence is derived from experimental studies with rather artificial settings, the relevance of these results in humans remains a matter of debate. Recently, it has also been proposed that HHcy and B-vitamin deficiencies may promote CVD through accelerated telomere shortening and telomere dysfunction. This review provides a critical overview of the existing literature regarding the role of HHcy and B-vitamin deficiencies in CVD. At present, the CVD risk associated with HHcy and B vitamins is not effectively actionable. Therefore, routine screening for HHcy in CVD patients is of limited value. However, B-vitamin depletion is rather common among the elderly, and in such cases existing deficiencies should be corrected. While Hcy-lowering with high doses of B vitamins has no beneficial effects in secondary CVD prevention, the role of Hcy in primary disease prevention is insufficiently studied. Therefore, more intervention and experimental studies are needed to address existing gaps in knowledge.

## 1. Introduction

Homocysteine (Hcy) is a sulfur-containing amino acid that is not a constituent of proteins. It is part of the intermediary metabolism and synthesized in the Hcy-methionine cycle by a multi-step process [1]. A high level of Hcy in the blood (hyperhomocysteinemia, HHcy) is correlated with numerous diseases, such as cardiovascular disease (CVD), renal dysfunction, bone fractures, and cognitive decline [2,3,4,5,6], see Figure 1. Upon adequate availability of methionine, Hcy is coupled to serine and subsequently catabolized to α-ketobutyrate and cysteine, a precursor of the principal antioxidant compound glutathione (GSH). This reaction is called transsulfuration and requires vitamin B6 (B6) as a co-factor for the enzymes cystathionine-β-synthase (CBS) and cystathionine-γ-lyase (CGL) [7]. In case the oral intake of methionine is low, Hcy is primarily transformed into methionine. This reaction is called remethylation and requires 5-methyltetrahydrofolate (5-MTHF, vitamin B9) as substrate and vitamin B12 (B12) as co-factor for the catalyzing enzyme methionine synthase (MS) [7]. Therefore, Hcy is also considered a functional marker for the availability of B9 and B12. Furthermore, it reflects the functionality of the B6-dependent transsulfuration pathway. In HHcy patients, toxic Hcy is eliminated through activation of the transsulfuration pathway and remethylation to S-adenosyl-methionine (SAM), the universal methyl-group donor required for virtually all methylation reactions [7]. Therefore, Hcy is an important modulator of the cell’s methylation capacity and oxidative status.

In humans, B12 is an essential cofactor for two enzymes: MS and methylmalonyl CoA mutase. MS is localized in the cytosol and catalyzes the remethylation of Hcy to methionine, whereas methylmalonyl CoA mutase is localized in the mitochondria where it transforms methylmalonyl coenzyme A (CoA) to succinyl CoA [7,8]. In B12 shortage, the activity of both enzymes is impaired leading to an accumulation of Hcy and methylmalonic acid (MMA) with increased plasma concentrations [9]. Although both metabolites are used as markers of B12 deficiency, MMA is considered to be more specific and sensitive than Hcy [9,10].

5-MTHF is the most abundant folate species in human serum accounting for 82–93% of the total folate [11,12]. It provides a methyl group for the remethylation of Hcy to form methionine and tetrahydrofolate (THF). The enzyme 5,10-methylenetetrahydrofolate reductase (MTHFR) converts 5,10-methylenetetrahydrofolate (5,10-MTHF into 5-MTHF and thus is crucial for the availability of 5-MTHF. The common MTHFR C677T polymorphism is associated with a reduced enzyme activity, which can limit the availability of 5-MTHF in situations when folate supply is low [13]. Recent evidence suggests that riboflavin in the form flavin adenine dinucleotide (FAD) is a cofactor for MTHFR and may thus represent a key factor linking the MTHFR C677T polymorphism to CVD [14].

Folate deficiency can cause neural tube defects [4] and HHcy [5]. Moreover, folate deficiency is associated with cardiovascular diseases, cognitive impairment, and cancer [2,3,5,6,8]. Elevated concentrations of total Hcy (tHcy) might be a risk factor or a marker for age-associated diseases [5,15,16,17]. The plasma and serum concentrations of total folate and tHcy depend on age. The age-related decrease in the concentration of folate [17] and the physiological decline in renal function explain the increase in tHcy levels in the elderly [17,18]. However, it is not known whether the methylated folate form that is the direct methyl donor for Hcy is present at a lower concentration in the elderly.

Upon adequate availability of methionine, Hcy is eliminated via the B6-dependent transsulfuration pathway. The term vitamin B6 actually represents a group of six water-soluble compounds with a central pyridine ring, which are involved as co-factors in numerous reactions, mainly in amino acid metabolism and fatty acid biosynthesis [19]. Pyridoxal 5′-phosphate (PLP) is the primary metabolically active form. B6 is found in a wide variety of food items including meat, fish, and fowl so that deficiencies are rare [20]. In addition, intestinal bacteria synthesize B6, which is mainly consumed by non-B6-synthesizing bacteria. 

## 2. Homocysteine, B-Vitamin Deficiency, and Mortality

Prospective clinical trials from many countries have shown that a high Hcy concentration in plasma or serum is a significant risk factor for mortality regardless of cause [6,15,16,18]. Among 17,361 participants of the Hordaland Homocysteine Study, Refsum et al. showed a significant increase in CVD and all-cause mortality, which was 3.6 times higher in subjects with Hcy ≥ 20 μmol/L compared to those with Hcy < 9 μmol/L [17]. A later meta-analysis of eleven prospective studies including 27,737 individuals (4110 deaths) showed a significant linear association between plasma Hcy and all-cause mortality [6]. The random effects model revealed a cumulative relative risk (RR) of 1.80 (95% CI: 1.51–2.14) for all-cause mortality in the highest Hcy category vs. the lowest category. The authors calculated a 33.6% higher risk for all-cause mortality for each 5 µmol/L increment in plasma Hcy (RR = 1.34; 95% CI: 1.25–1.42). Our own results from the Ludwigshafen Risk and Cardiovascular Health (LURIC) study identified plasma Hcy as an independent predictor of mortality that is also inversely associated with leucocyte telomere length (LTL) [1]. Vollset et al. reported a 49% increase in all-cause mortality per 5 µmol/L increment of plasma Hcy [18]. Despite robust evidence that links plasma HCY to mortality risk, not all existing studies are consistent. In the Longitudinal Aging Study Amsterdam (LASA), Swart et al. found elevated plasma Hcy to be associated with mortality in older women, but not men [21]. In line with this observation, a recent meta-analysis of six studies showed an increased risk of all-cause mortality in women (RR 1.74; 95% CI: 1.24–2.44, *p* = 0.001) and in the whole population (RR 1.93; 95% CI: 1.54–2.43, *p* < 0.001), but not in men (RR 1.87; 95% CI: 0.64–5.50, *p* = 0.255) [12]. Moreover, this meta-analysis revealed a pooled RR of all-cause mortality of 1.27 (95% CI: 1.03–1.55, *p* = 0.023) per 5 μmol/L increment in circulating Hcy in the whole population. However, caution is warranted when interpreting potential differences between men and women in this meta-analysis. Only two out of the six included studies were used for sex comparisons with very different results for men, but not women. Therefore, existing studies are insufficient to draw firm conclusions on gender differences. In addition to all-cause mortality, elevated plasma Hcy is also related to an increased risk of CVD [22,23]. Despite high Hcy being a generally accepted risk factor for mortality [6], only a few large clinical trials have studied the correlation between plasma B6 and mortality [1,24,25]. In the LURIC cohort, we found a strong inverse relationship between plasma B6 and mortality [1]. Subjects with B6 concentrations above 14.2 μg/L showed a 59% lower all-cause or CVD mortality when compared to those with lower B6 concentrations. Patterson et al. studied 28 biomarkers in middle-aged men from the Caerphilly Prospective Study (CaPS study), and found an inverse relationship between B6 and non-CVD mortality (HR 0.83, 95% CI: 0.75–0.93 *p* < 0.01) [25]. However, this association was not present for CVD mortality. Additionally, in another cohort of 7796 patients with stable angina pectoris or MI, plasma B6 predicted all-cause mortality in MI patients, after adjustment for common confounders [24]. However, when including inflammatory markers in the Cox-regression model, this association disappeared [24]. The authors suggested that decreased B6 concentrations in plasma could be secondary to inflammatory activation. They hypothesized that inflammation instead of low B6 is responsible for the correlation with all-cause mortality [24]. Contrarily, in LURIC, B6 was identified as an independent predictor of all-cause mortality after adjustment for high-sensitive C-reactive protein (hs-CRP), interleukine-6 (IL-6), and other established risk factors [1].

HHcy caused by B12 deficiency has also been linked to mortality [1,6,18,26,27,28]. B12 deficiency is common in elderly [29], hospitalized [30], and ill individuals [31]. The main causes of B12 deficiency in elderly individuals are food-cobalamin malabsorption (60–70%) and pernicious anemia (15–20%) [32]. Food-cobalamin malabsorption syndrome refers to the inability to release B12 from food or from intestinal transport proteins in the presence of hypochlorhydria and is frequently the result of gastric atrophy [32,33]. A reduced dietary intake rarely leads to B12 deficiency (<5%) and is mainly a concern in vegetarian and vegan individuals [7]. Cobalamin deficiency is a slowly progressing process with different stages [7]. In the early stages, plasma and cells become depleted of the vitamin causing lowered serum concentration of holotranscobalamin (holoTC) but Hcy and MMA are still within the reference range. If the negative balance continuous, the metabolic markers Hcy and MMA start to rise so that elevated concentrations can be measured in serum. Pernicious anemia characterized by antibodies against intrinsic factors is the most widely known disorder of cobalamin absorption and is very frequent among elderly patients (about 50% of the cases). Aging and weakness are associated with B12 deficiency independently of nutritional intake [34]. B12 deficiency is linked with an increased incidence of age-related conditions, such as neurodegenerative disease, coronary heart disease (CHD), and osteoporosis [29,35]. However, evidence that links B12 to mortality is largely lacking. So far, only from LURIC has a higher mortality been reported in patients with the lowest plasma B12 concentrations compared to those in the mid-range [27]. 

Surprisingly, a high B12 plasma concentration is also correlated with higher mortality of all causes [36,37,38,39]. However, after correction for hepatic function this correlation is no longer significant [39]. It has been hypothesized that liver disease with an impaired integrity of hepatocytes and reduced liver function can cause elevated plasma B12 concentrations by a release of stored B12 from lytic hepatocytes (in the form of holohaptocorrin), reduced hepatic transcobalamin II synthesis (limits intestinal B12 uptake), and impaired hepatic uptake of holotranscobalamin II [39]. Despite elevated B12 having been correlated to higher mortality in some clinical trials [36,37,38,39], this association is not generally established. In 2239 critically ill patients, elevated plasma B12 (>1593 pmol/L) was associated with an increased 90-day mortality [36]. In the Newcastle 85+ study, subjects with B12 > 500 pmol/L had a 40% higher risk of all-cause mortality compared to those with B12 < 500 pmol/L (HR1.41, 95% CI: 1.02–1.95, *p* = 0.039) [37]. In another study of heart failure patients, B12 concentrations above 270 pg/mL had 80% sensitivity and 58% specificity for predicting all-cause mortality (area under the curve = 0.67, 95% CI: 0.56–0.78, *p* = 0.003) [30]. A large observational study by Callaghan et al. showed unadjusted odds ratios of 2.83 (95% CI: 2.13–3.76) and 2.72 (95% CI: 2.08–3.55) for 30-day and 90-day death, respectively, in subjects with B12 concentration >1000 pmol/L [39]. However, currently available results from clinical trials indicate that the abnormal liver function itself rather than a high B12 concentration in plasma push mortality in this context. In addition, it needs to be mentioned that elevated B12 plasma concentrations are significantly associated with hematologic malignancies and solid cancers. A large comparison study of individuals with total plasma B12 concentrations above and below 1000 pg/mL demonstrated a strong association of high B12 with solid organ cancer, which was strongest in the presence of metastases [40]. 

## 3. HCY, B-Vitamins, and CVD Risk

Plasma HCY is an established risk factor for CVD and stroke [41]. This has been shown by numerous retrospective and prospective observation studies and is further substantiated by several meta-analyses [42,43]. Interestingly, a meta-analysis by the Homocysteine Studies Collaboration [42] found stronger associations between plasma Hcy and CVD including stroke in retrospective studies than in prospective studies. This meta-analysis included 5073 heart disease events and 1113 stroke events. After adjustment for known cardiovascular risk factors and regression dilution bias, it was demonstrated that a 25% lower than usual plasma Hcy concentration (approximately 3 μmol/L lower) is associated with an 11% lower CVD risk (OR 0.89; 95% CI: 0.83–0.96) and a 19% lower risk of stroke (OR 0.81; 95% CI: 0.69–0.95). In a meta-analysis of prospective studies only, Wald et al. showed similar results, with an estimated risk reduction per 3 μmol/L lower plasma Hcy of −16% (95% CI: 11–20) for heart disease and −24% (95% CI: 15–33) for stroke [43]. A polymorphism in position 677 of the MTHFR gene has been identified as genetic risk factor for HHcy. The substitution of cytosine (C) by thymine (T) results in an amino acid change in alanine to valine. Affected individuals have a reduced MTHFR enzyme activity with higher plasma Hcy concentrations than those with the wild-type genotype [44]. The TT genotype is rather common with an average prevalence of 12% [45]. In a meta-analysis of observational case-control studies with a total of 11,162 cases and 12,758 controls, Klerk et al. [46] estimated that individuals with the MTHFR 677TT genotype have 16% higher odds of CVD (OR 1.16; 95% CI: 1.05–1.28) compared to individuals with the CC genotype. Similar results were reported in a meta-analysis of 75 studies with 22,068 cases and 23,618 controls [47]. TT homozygotes had an odds ratio of 1.16 (95% CI: 1.04–1.29) for ischemic heart disease than CC homozygotes and a 1.9 µmol/L higher average plasma Hcy concentration. According to our current understanding, the MTHFR C677T gene mutation only becomes relevant when either folate or riboflavin (vitamin B2) status is low. In folate deficient individuals with the MTHFR C677T polymorphism, residual enzyme activity is insufficient to provide enough methyl groups for the remethylation of HCY. Even when folate availability is sufficient, deficiency of, another co-factor of MTHFR, can further reduce the residual MTHFR activity and thus limit the production of 5-MTHF. In this context, B2 deficiency has been found to increase the genetic risk of hypertension in individuals with the TT genotype [14,48]. However, remethylation of HCY is only of importance when methionine uptake via the food is low. In LURIC, individuals with a Hcy concentration ≥ 9.8 μmol/L had a 28% higher risk to die from CVD or any other cause compared to those with a lower Hcy concentration [1]. Existing studies suggest that HCY promotes CVD risk through adverse effects on vascular smooth muscle cells, endothelial function, lipid peroxidation, and thrombogenesis [49,50,51]. 

Although a well-balanced diet caters for most nutritional requirements, vitamin supplements are widely used around the globe. As the supplementation of B vitamins (B6, B9, and B12) effectively reduces HCY, it was hypothesized that the regular administration of these vitamins reduces CVD risk [52]. However, evidence from large randomized controlled trials over the last 2 decades does not support this idea [53,54,55,56,57,58,59,60,61,62]. The Women’s Antioxidant and Folic Acid Cardiovascular Study (WAFACS) [53], a double-blind RCT that administered placebo or a combination of B6, B9, and B12 to women with CVD or ≥3 coronary risk factors, demonstrated no reduction in major CVD events despite a significant lowering of plasma Hcy. Similar results were obtained in the Heart Outcomes Prevention Evaluation-2 (HOPE-2) trial with no reduction in CVD mortality despite reduction in Hcy concentrations in patients with diabetes or vascular disease [54]. In addition, in the Western Norway B Vitamin Intervention Trial (WENBIT), a secondary prevention trial that enrolled patients undergoing coronary angiography, treatment with B12 and B9 with or without vitamin B6 failed to show an effect on total mortality or CVD events [55]. Other intervention studies that did not find beneficial effects of B-vitamin supplementation on CVD mortality include the Norwegian Vitamin (NORVIT) [56], and the Vitamin Intervention for Stroke Prevention (VISP) trials [57]. 

In contrast to the lacking effect on CVD mortality, the HOPE-2 trial showed a 25% risk reduction in stroke during 5 years of B vitamin supplementation [54]. However, this result could not be confirmed in the VISP trial [57]. The Vitamins to Prevent Stroke (VITATOPS) trial investigated the utility of B-vitamin supplementation (B6, B9, and B12) for secondary disease prevention in patients with recent stroke or transient ischemic events, but failed to show any reduction in major vascular outcomes [58]. Similar findings were obtained in the Effectiveness of Additional Reductions in Cholesterol and Homocysteine (SEARCH) trial that included cardiovascular patients with prior myocardial infarction [59]. Other treatment studies also failed to demonstrate beneficial effects of B6, B9, and B12 administration in CVD patients [60,61,62], hemodialysis patients [63], and kidney transplant recipients [64]. In line with the lacking effect on vascular endpoints, a meta-analysis of 12 B-vitamin supplementation studies in subjects with pre-existing vascular diseases showed an improved flow-mediated vasodilation (a surrogate marker of vascular health) in the short term (< 8 weeks), but not in the long term [65]. Other meta-analyses and Cochrane reviews reached a similar conclusion that B-vitamin supplementation has no significant effect on CVD risk despite a lowering of plasma Hcy [66,67]. Interestingly, meta-analyses by Huang et al. and Yi et al. reported that the oral intake of B vitamin lowers the risk of stroke, but does not improve the risk of CVD, myocardial infarction (MI), CHD, cardiovascular death, or mortality of any-cause [66,68].

While most primary and secondary prevention studies showed no reduction in B vitamin supplementation on CVD and stroke events, the China Stroke Primary Prevention Trial (CSPPT) demonstrated a 21% risk reduction in first stroke when treating hypertensive adults without history of stroke or MI for 5 years with a combination of enalapril (daily 10 mg) and B9 (folic acid 0.8 mg daily) rather than enalapril alone [69]. This double-blind RCT enrolled 20,702 adults. Furthermore, folic acid fortification of enriched grain products (140 μg folic acid per 100 g flour, implemented in 1998 in the US and Canada, resulted in a population-wide reduction in plasma Hcy, which was expected to reduce stroke mortality [70]. The decrease in stroke mortality in US was estimated −0.3 resp. −1.0% per year from 1990 to 1997 and accelerated to −2.9 resp. −5.4% per year in 1998 to 2002. In contrast, the decline in stroke mortality in England and Wales without folic acid fortification did not change significantly between 1990 and 2002. Based on these results it has been concluded that B9 might have some beneficial effects in primary prevention.

When supplemented in small doses, many vitamins are safe, but long-term consumption of mega-doses may cause harm [71,72]. There is some evidence that the arbitrary and uncontrolled use of vitamin preparations might be harmful. For example, amongst 38,772 older women (mean age, 61.6 years) who participated in the Iowa Women’s Health Study users of multivitamins, B6, B9, or minerals had an increased risk of mortality when compared with nonusers [72,73]. This study points out that the self-initiated, uncontrolled intake of vitamin or mineral supplements in elderly women could be harmful [72]. Because there is no evidence to support the use of Hcy-lowering supplements to reduce CVD events, the American Heart Association does not recommend the routine use of B vitamins for CVD risk reduction [74]. Only patients who are deficient or at risk for vitamin deficiency, such as those with malabsorption, total parenteral nutrition, or poor nutritional status, and specific subgroups (pregnant women) should receive an appropriate supplementation. 

## 4. Hyperhomocysteinemia and Vascular Disease—Mechanistic Aspects

To date, the mechanisms that mediate the adverse effects of HHcy are only partly understood. According to our current understanding HHcy increases oxidative stress, which modulates a number of downstream signaling pathways [75], see Figure 2. Furthermore, HHcy induces endoplasmatic reticulum (ER) stress [75,76]. Through alterations of DNA methylation HHcy also influences the expression of numerous pathogenic and protective genes. Moreover, Hcy can bind directly to proteins via its reactive sulfhydryl-group. This homocysteinylation of proteins can change protein function and alter the intracellular redox state.

Oxidative stress refers to the accumulation of reactive oxygen species (ROS), such as superoxide (O_2_^−^), hydrogen peroxide (H_2_O_2_), or peroxynitrite. Hcy stimulates the production of superoxide and peroxynitrite in circulating leucocytes and resident cells of the vessel wall by upregulating nicotinamide adenine dinucleotide phosphate (NADPH) oxidase (NOX) expression and the activation of calcium signaling with subsequent mitochondrial dysfunction [76,77,78,79]. NOX refers to a group of electron-transporting membrane proteins that are involved in the generation of O_2_^−^ and H_2_O_2_ [80]. Superoxide dismutase catalyzes the conversion of superoxide to H_2_O_2_. Moreover, superoxide induces the production of peroxynitrite by up-regulating inducible nitric oxide synthase (iNOS) and the uncoupling of endothelial nitric oxide synthase (eNOS). In addition to an increased ROS synthesis, Hcy also decreases the availability of antioxidants, such as haem-oxygenase-1 (HO-1), thioredoxin, and catalase [76,77]. Furthermore, Hcy inhibits eNOS-mediated NO synthesis and upregulates iNOS, which stimulates peroxynitrite production [76,81].

Hcy-induced oxidative stress triggers inflammatory processes mediated by NF-κB (nuclear factor ‘kappa-light-chain-enhancer’ of activated B-cells), endothelial dysfunction, adventitial activation, and the formation of the neutrophil extracellular trap [76]. In vitro and in vivo studies have shown that Hcy-induced oxidative stress induces the production and secretion of immunomodulators, such as CRP, IL-1β, IL-6, IL-8, and tumor necrosis factor alpha (TNFα) [1,77,82,83,84,85]. In addition, patients with HHcy have increased concentrations of soluble CD154, also called sCD40L, a protein of the TNF superfamily, which, upon binding to its receptor CD40 on the surface of antigen-presenting cells, mediates a variety of immune and inflammatory responses [86,87]. In addition, Hcy promotes B-lymphocyte proliferation and immunoglobulin G (IgG) secretion [78,88]. Moreover, oxidative stress impairs endothelial vascular function through reduced NO synthesis and vascular smooth muscle cell de-differentiation. ROS inhibit the activity of dimethylarginine dimethylaminohydrolase resulting in an accumulation of asymmetric dimethylargine (ADMA), a competitor of L-arginine for eNOS. Together with a reduced availability of L-arginine, this causes eNOS uncoupling [76].

The adventitia of blood vessels is another critical target for the adverse oxidative effects of Hcy. Hcy-induced NOX activation increases ROS production and the recruitment of inflammatory cells. In vivo studies have shown that HHcy stimulates macrophage infiltration, IL-6, and CC-chemokine ligand 2 (CCL-2; synonym: monocyte chemotactic protein-1) production and matrix metalloproteinase (MMP) proteolytic activity in the aortic adventitia. This causes a weaking weaking of the aortic wall with an increased susceptibility for aortic aneurysm and aortic dissection [89]. Furthermore, HHcy promotes adventitial hyperplasia and collagen I deposition [90]. 

A rather novel mechanism that may also be involved in the adverse oxidative effects of Hcy in the vessel wall is the formation of neutrophil traps, a network of fibers composed of neutrophil DNA and proteins, which eliminate pathogenic components without phagocytosis [91]. Such traps have been described in hyperhomocysteinemic type 2 diabetic patients and are considered an inflammatory neutrophil activation. The formation of neutrophil traps is triggered by NOX-derived ROS that promote the migration of neutrophils to the vessel wall [92]. Another route that mediates Hcy-induced oxidative stress is the direct binding of Hcy to *N*-Methyl-D-Aspartate (NMDA) and other receptors on the cell surface. Several in vitro experiments have shown that upon activation these receptors increase calcium signaling and induce mitochondrial dysfunction, with subsequent ROS formation and expression of cyclooxygenase 2 (COX-2) [93,94]. Activation of cell surface receptors by Hcy also regulates the translocation of ß-catenin to the nucleus, which alters the expression of claudin-5 and disrupts endothelial cell junctions [94].

The ER, where newly formed proteins undergo post-translational modifications and assume their definitive confirmation, is another target of Hcy-induced oxidative stress. Misfolded proteins are either re-folded or degraded by the ER. HHcy can increase the quantity of misfolded proteins, which activates compensatory mechanisms in the ER and may cause an accumulation of unfolded and aggregated proteins in the ER of endothelial cells, vascular smooth muscle cells, and lymphocytes [76,95,96,97]. Such ER dysfunction triggers the expression of ER stress response genes, such as glucose-regulated protein 78 (GRP78), activating transcription factor 6 (ATF6), protein kinase RNA-like endoplasmatic reticulum kinase (PERK), and others. The activation of ER stress response genes promotes the development of atherosclerosis through endothelial dysfunction, apoptosis and cell death, inflammation, and T-cell activation in the vessel wall. There is also evidence that Hcy-induced ER stress and oxidative stress reinforce each other [76]. 

In addition to oxidative stress related pathogenic effects, Hcy also modulates the expression of multiple genes by changes in DNA methylation [98]. An impaired activity of methionine synthase and/or MTHFR due to B9 and B12 deficiency or genetic variants causes accumulation of S-adenosyl homocysteine (SAH) and decreases the intracellular S-adenosyl methionine (SAM) level) [99]. This results in a reduced SAM/SAH ratio and leads to global hypomethylation. DNA hypomethylation in particular has far reaching consequences as it modifies the expression of many genes [98,100]. For example, the increased expression of telomerase and p21 in response to Hcy-induced hypomethylation contributes to accelerated telomere shortening and senescence in endothelial cells [101]. In vascular smooth muscle cells, the up-regulation of platelet-derived growth factor (PDGF) stimulates proliferation and migration, which leads to neointimal formation [102]. SAM can counteract neointima formation by reducing ER stress and inflammation. The methylation of cytosine bases in eukaryotic DNA by DNA methyl transferase (DNMT) results in the formation of 5-methylcytosine nucleotides, which are typically located adjacent to a guanine nucleotide forming so called CpG islands [103,104]. The methylated cytosine impairs the interaction of transcription factors with many DNA promotors as the majority of them is located within such CpG islands. Depending on the type of transcription factor, this can result in a repression or up-regulation of gene expression. Another result of DNA methylation is the deacetylation of histone H4 and an altered methylation of histone H3, which modifies chromatin structure [105]. In contrast to DNA, histones are methylated on the side chains of lysines and arginines through the action of histone methyl transferases. The potential relevance of histone methylation in vascular disease is supported by a study from Cong et al. showing that the inhibition of histone methylation enhances macrophage apoptosis [106]. Hcy not only reduces DNA methylation through a decreased SAM/SAH ratio, but also modulates DNMT expression. Interestingly, in vascular cells Hcy can inhibit or activate DNMT resulting in an increased expression of pathogenic genes and a reduced expression of protective genes. In addition, messenger RNA (mRNA) methylation seems to be involved in the epigenetic effects of Hcy that mediate atherosclerosis. In summary, there is substantial evidence that Hcy modifies genes expression in blood vessels via DNA, RNA, and histone methylation. 

Lastly, Hcy can bind directly to proteins, thus altering their structure and function. Homocysteinylation can occur via the thiol group (S-homocysteinylation) or the amino group (*N*-homocysteinylation) of homocysteine [107]. During S-homocysteinylation, the free thiol group of HCY binds to another free thiol group of a cysteine residue in the target protein forming a disulfide bond. This reaction is reversible so that Hcy can also be detached from the target protein. In contrast, *N*-homocysteinylation occurs when Hcy thiolactone, the cyclic thioester of Hcy, binds irreversibly via its amino group to the amino group of a lysin residue in the target protein. This reaction opens the thiolactone ring structure, which results in the formation of a new free thiol group that can modify the redox status. An increased protein homocysteinylation has been reported in HHcy patients with diabetes and atherosclerosis. Fibronectin, eNOS, and angiotensin-converting enzyme (ACE) are a few examples where homocysteinylation has been reported to modify biochemical properties. In addition, paraoxonase 1 (PON1), a calcium-dependent multifunctional enzyme contained in high-density lipoprotein (HDL) particles, seems to mitigate the pathological effects of Hcy and in particular of Hcy thiolactone [108,109,110]. Hcy thiolactone is a naturally occurring, particularly atherogenic form of Hcy that arises when Hcy is mistakenly selected by methionyl-tRNA synthetase during protein synthesis. PON1 hydrolyzes aryl esters, lactones, and organophosphates. Due to this function, PON1 detoxifies Hcy thiolactone and thus protects against *N*-homocysteinylation of proteins. Ultimately, PON1 is an important mediator of the antioxidant properties of HDL, which protect low-density lipoprotein from oxidation and thus limits vascular subendothelial damage.

All mechanisms mentioned so far suggest that Hcy actively promotes vascular disease via direct and indirect pathways. However, results from in vitro experiments demonstrate that activation of the immune system can raise the plasma Hcy concentration making it more an indicator than an actor. For example, IL-1β and TNFα have been shown to alter the cells’ redox state and to increase the extracellular Hcy concentration in a concentration-dependent fashion [107]. In line with this concept, the administration of a combination of B6, B9, and B12 for a period of 7.3 years did not improve hsCRP, IL-6, and markers of endothelial impairment [111]. In addition, other studies that aimed to reduce plasma Hcy by oral intake of B-vitamin preparations did not lower markers of inflammation in blood plasma [111,112,113]. Moreover, systemic inflammation has been found to increase B6 catabolism and cellular uptake, leading to low B6 concentrations in plasma [114]. Conversely, a clinical trial by Ulvik et al. showed that B6 supplementation lowers systemic inflammation in patients with stable angina pectoris [115]. It can be speculated that B6 represents an important link between Hcy, the cells’ redox state, and systemic inflammation. However, the interactions between Hcy, B vitamins, and the immune system are complex [116], and existing results are still controversial [111,117,118]. 

In summary, Hcy interferes with multiple pathways and mechanisms that are involved in atherosclerosis and vascular disease. However, the relevance of these mechanisms in CVD is insufficiently understood. Many controversies remain because the respective results are mainly derived from in vitro experiments with a rather artificial setting. Cell culture studies often used Hcy concentrations that are outside the physiological range in humans. Therefore, more specific in vivo studies are needed to further explore how Hcy drives or modifies CVD. 

## 5. HHcy, B-Vitamin Deficiency and Genomic Effects, and Telomere Length 

In addition to CVD, several other chronic illnesses of advanced age, such as dementia, osteoporosis, and autoimmune disease have also been linked to HHcy and B vitamin deficiency [119]. The association with age-related diseases has sparked speculations about adverse effects of HHcy and B-vitamin deficiencies on telomere function. Telomeres are protective nucleoprotein structures at the ends of eukaryotic chromosomes that are of critical importance for the preservation of our genome [120,121]. Because of their progressive shortening, telomeres are considered a molecular clock of aging. With every cell division, telomeres shorten a little bit due to incomplete replication of the DNA lagging strand. In addition, accidental damage can also cause telomere shortening. This age-related shortening progressively compromises telomere function and triggers senescence [120,121,122]. Prospective observational studies in healthy and high-risk populations could demonstrate that short telomeres substantially increase the risk of all-cause and cardiovascular mortality [120,122,123]. For example, in the LURIC study, patients in the upper three quartiles of RTL had a lower HR for all-cause (HR: 0.822; 95% CI: 0.71–0.92, *p* = 0.008) and CVD mortality (HR: 0.836; 95% CI: 0.72–0.97, *p* = 0.017) when compared to those in the first quartile with short telomeres [123]. Not only short telomeres, but also dysfunctional telomeres are related to increased mortality [107,108,110]. The role of telomeres and telomerase in CVD has recently been reviewed by Boniewska-Bernacka et al. [124].

Environmental and lifestyle factors can influence the progression of telomere shortening and promote telomere dysfunction [119]. Most of these factors are correlated with higher levels of ROS and oxidative stress, which can cause DNA base damage, DNA strand breaks, and accelerated telomere shortening [125]. Previous studies have repeatedly proposed a mechanistic relationship between accelerated telomere attrition, oxidative stress, and systemic inflammation in individuals with HHcy and B vitamin deficiency [1,125,126,127]. However, existing studies are inconsistent. An inverse correlation between telomere length (TL) and Hcy was observed in some clinical trials [128,129,130] but not in all [131,132,133,134]. Amongst 1319 individuals from a population-based cohort, Richards et al. showed a difference in LTL between the third and first tertile of plasma Hcy that corresponded to 6 years of telomeric aging [128]. Such a difference may, at least partly, be due to a lower expression of human telomerase reverse transcriptase (hTERT) and DNA hypomethylation [98,135]. A causal link between Hcy and telomeres has also been demonstrated in animals, where LTL, TERT mRNA expression, and methylation of the TERT promotor region were lower in hyperhomocysteinemic animals than in controls [136]. Besides telomere shortening and DNA hypomethylation, HHcy also promotes other types of genomic damage, such as the development of micronuclei and DNA interstrand cross-links [136,137]. In LURIC, subjects with shorter telomeres were characterized by higher levels of Hcy, lower B6, but higher IL-6 and CRP concentrations [1]. These subjects also had the highest mortality. In addition, the large-scale National Health and Nutrition Examination Survey (NHANES) showed a positive association between B6 intake and LTL [126]. However, other studies did not find a significant correlation between LTL and plasma B6 or B6 intake [131,132,133]. 

Available data point towards oxidative stress and chronic inflammation as key mediators that link Hcy, B6, and LTL with mortality. Hcy compromises enzymatic and non-enzymatic antioxidant defense mechanisms in many tissues [111,138,139] and the resulting oxidative stress triggers inflammation through intra- and extracellular damage [140]. In LURIC, for example, Hcy and IL-6 were the strongest determinants of LTL. Experimental data indicate that oxidative DNA damage impairs recognition and binding of the shelterin proteins telomeric repeat binding factor 1 (TRF1) and telomeric repeat binding factor 2 (TRF2) to telomeric DNA [141]. These proteins are essential for the three-dimensional structure of telomeres and their correct function. Furthermore, ROS promote 8-oxoguanine formation, a common DNA damage that causes mismatch pairing with adenine leading to G-to-T and C-to-A substitutions in the genome [125]. Finally, ROS also impair the repair of oxidative DNA damage via a decrease in endonuclease III-like protein 1, which recognizes and corrects base damage of pyrimidines [142]. In summary, the different types of oxidative DNA damage compromise the protective function of telomeres, trigger systemic inflammation, and drive cellular senescence through a senescence-associated secretory phenotype (SASP) [143]. 

An insufficient supply with B12 is also believed to increase mortality through HHcy and accelerate telomere shortening [6,18,27,120,121,122]. Through its role in the folate cycle, which provides methyl groups for DNA and histone methylation, B12 is important for genomic stability [7,144]. Similar to B6, a low supply of B12 can also disturb the subtle intracellular equilibrium between oxidants and anti-oxidants [145]. So far, the relationship between TL and B12 has not been studied comprehensively. Our own results from the LURIC study support a link between Hcy, TL, and mortality in individuals with low plasma B12 [1,27]. Individuals with pronounced functional B12 deficiency appear to develop a pro-oxidative environment with HHcy that facilitates hypomethylation and DNA damage, which ultimately increases all cause and CVD mortality [26,120,121,122]. Interestingly, not only low, but also high B12 plasma concentrations are associated with mortality [27]. Results from an own study suggest that systemic inflammation results in a release of B12 from intracellular stores, accelerates telomere shortening, and ultimately increases mortality [27]. However, it has to be mentioned that some smaller studies showed no significant associations between plasma B12 and TL [130,132,133,134]. 

## 6. Controversial Aspects of Hcy and B Vitamins in CVD

While countless studies have firmly established Hcy as biomarker for CVD risk, more than two decades of intensive research activities have failed to demonstrate that Hcy lowering through B-vitamin supplementation results in a net reduction in CVD risk. This unresolved conundrum has raised doubts about a causal involvement of HHcy and B-vitamin deficiencies in atherosclerosis. In this context, it is important to understand that a variable may predict a disease without being causally involved in its pathogenesis. Therefore, the modification of such a variable does not alter disease incidence, unless it also targets a causal mechanism. For example, lowering glycated hemoglobin A1c (HbA1c) without improving glycemic control would not reduce diabetic complications. For several reasons, available evidence is insufficient to rule out a mechanistic involvement of Hcy and B vitamins in atherosclerosis [146]. Although existing intervention studies have consistently shown that high-dose B-vitamin supplementation does not reduce CVD incidence or progression, their nature does not rule out a causal relationship. One important factor is the duration of intervention studies. Whereas the differences in observational studies have developed over very long intervals (decades), most intervention studies lasted for less than 5 years. As atherosclerotic plaque formation takes decades, it is probably unreasonable to expect measurable beneficial effects of intervention studies in such a short time. This argument is supported by a meta-analysis that showed a risk reduction for stroke after 36 months of folate supplementation, but not earlier [147]. The potential for favorable effects in previous intervention studies could have been reduced further by relatively normal Hcy baseline concentrations amongst the participants, which ranged well below 20 µmol/L on average. Furthermore, the meta-analysis by Wang et al. found no effect in populations with folate fortification of grain and when Hcy was lowered by less than 20%. Finally, most CVD patients are treated with disease specific medication. The effects of additional B vitamin supplementation may simply be too weak to induce a further improvement in clinical outcome. 

When interpreting the results of existing intervention studies, the possibility of dual effects has largely been ignored. The absence of a significant net-effect could simply be the sum of more than one effect that oppose each other. While the reduction in Hcy might positively influence the cells redox state, this effect could be offset by adverse inflammatory and proliferative effects. Such adverse effects have been postulated for unmetabolized folic acid, which becomes increasingly available when high doses of folic acid are supplemented. Furthermore, the propagation of neoangiogenesis and smooth muscle cell proliferation in atherosclerosis depend upon B9 and B12 availability as these compounds nurture essential transmethylation reactions and DNA/RNA synthesis. Therefore, the efficacy of B-vitamin supplementation in CVD may depend on disease stage. In disease free individuals, Hcy-lowering may be beneficial due to the reduction in oxidative stress, the preservation of telomeres and epigenetic patterns. However, once a plaque has been formed, the beneficial effects of Hcy-lowering may be outweighed by the potential adverse effects of B vitamins on proliferation and inflammation. Furthermore, DNA damage and premature telomere shortening may not be reversible by B-vitamin supplementation, and consequently, downstream effects, such as altered gene expression pattern, remain unaffected. Some evidence for disease-stage specific effects of B vitamins comes from the NHANES study showing that females between 35 and 55 years with high B9 plasma concentrations have a low risk of incident CVD, whereas older women had the lowest risk when plasma folate B9 was low [148]. Moreover, folic acid supplementation for primary prevention has been found to reduce significantly incident stroke events [69]. In the large Chinese stroke prevention study, where 20,000 healthy individuals received either folic acid or placebo, the active treatment group showed a significant reduction in first stroke HR: 0.79 (95% CI: 0.68–0.93). However, the different genetic background in the Chinese population with a high frequency of MTHFR 677C to T polymorphism might have contributed to the study outcome. Moreover, the antifolate methotrexate has been suggested to reduce CVD risk in rheumatoid arthritis patients beyond the effect of other ant-inflammatory agents [149]. However, a large randomized intervention trial including 4786 CVD patients, failed to show a reduction in CVD events after more than two years of low-dose methotrexate treatment [150]. Some additional evidence that supports a causal role of B vitamins in the early stages of atherosclerosis comes from a couple of animal studies [151,152,153]. Finally, B-vitamin supplementation alone may be insufficient to modify CVD risk, because it does not change DNA methylation patterns and does not protect telomeres. This may be particularly relevant in individuals with pre-existing CVD. Additionally, it cannot be ruled out that additional micronutrients, such as n-3 polyunsaturated fatty acids (PUFA), olive oil, vitamin C and E, polyphenols, legumes, and others have to be combined with B vitamins in order to obtain a measurable effect [154]. This concept is supported by treatment studies investigating the relationship of Mediterranean diet on incidence/mortality of CVD and stroke [154,155]. A meta-analysis by Rosato et al. showed a pooled RR of 0.70 (95% CI: 0.62–0.80) for CHD/AMI, and of 0.73 (95% CI: 0.59–0.91) for stroke when comparing individuals with the highest versus the lowest Mediterranean diet adherence score. Furthermore, Tsoukalas et al. studied the effects of a combination of nutraceutical supplements containing B vitamins, n-3 PUFA, vitamins C and E, and others on TL in healthy volunteers, and found an increase of TL of the whole telomere genome [156]. 

## 7. Conclusions

While plasma Hcy is an established indicator of CVD, it is still unclear whether or not it is causally involved in the pathogenesis of atherosclerosis. Therefore, routine screening for HHcy in CVD patients is not meaningful because this condition is currently not actionable. In particular, the administration of high doses of B6, B9, and B12 does not reduce CVD events within a reasonable time. Only in individuals with the MTHFR 677TT genotype or if B-vitamin depletion or where an inborn error of metabolism is suspected, should Hcy, B9, and B12 be measured, and supplementation considered. Of note, B-vitamin depletion is rather common among the elderly, and in such cases existing deficiencies should be corrected. However, there is no evidence that the associated reduction in Hcy has a favorable impact on CVD risk. In contrast to B-vitamin supplementation for secondary CVD prevention, epidemiologic data from some countries with prophylactic folic acid enrichment of grain products and results from large primary prevention studies show a lowering of primary stroke events and a decrease in neural tube defects. 

More work is needed to better understand the potential adverse effects of Hcy on blood vessels and how they can be mitigated. Another important point is the systematic investigation of the potentially adverse effects of individual vitamin species. Through additional primary prevention studies and the evaluation of epidemiologic data, the theory that Hcy and B vitamins are of particular relevance during early disease stages should further be explored. Finally, last but not least, longer and better-powered intervention studies should be performed to address the open questions from previous studies. 

## Figures and Tables

**Figure 1 nutrients-14-01412-f001:**
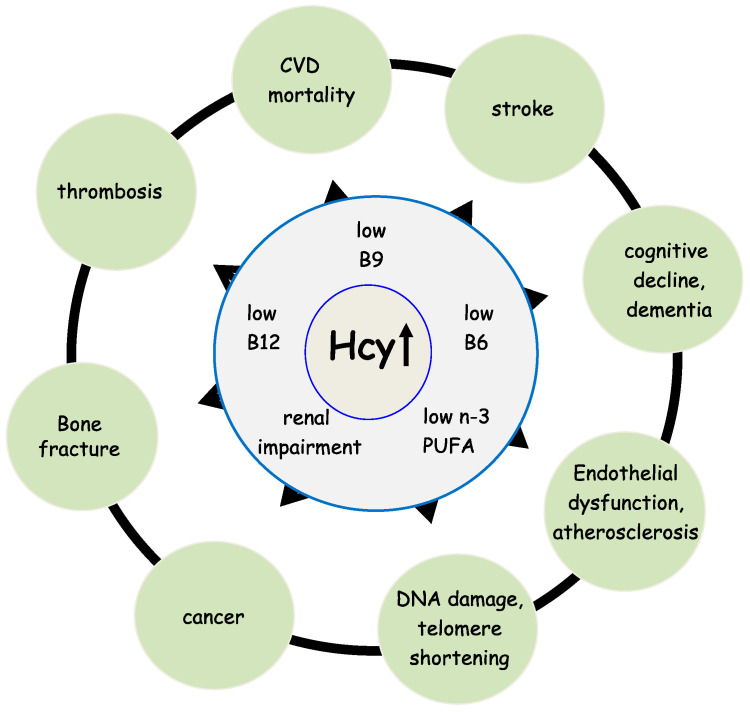
Hyperhomocysteinemia, B vitamins deficiency and age-related diseases. Hcy, homocysteine; PUFA, polyunsaturated fatty acids; ↑ increase.

**Figure 2 nutrients-14-01412-f002:**
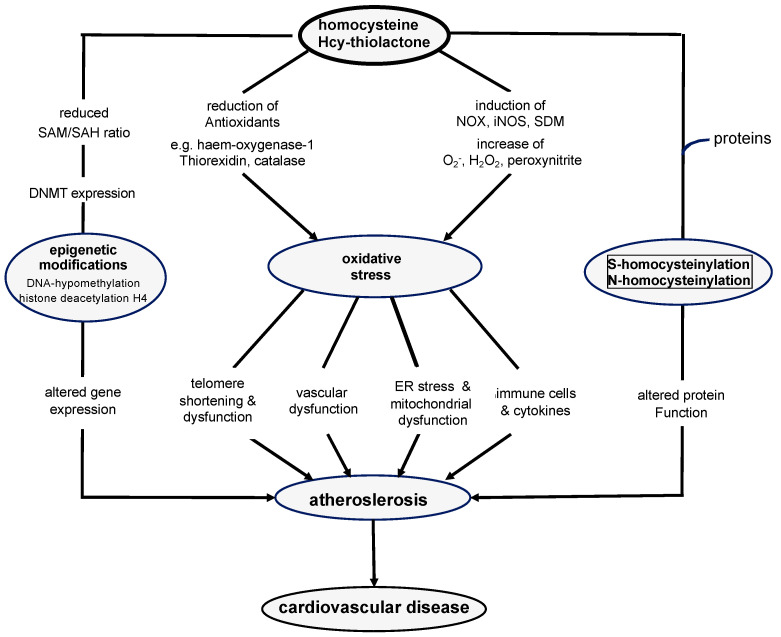
Pathomechanisms in hyperhomocysteinemia—oxidative stress, epigenetic modifications, altered protein function—causing atherosclerosis and cardiovascular disease. Hcy, homocysteine; SAH, S-adenosylhomocysteine; SAM, S-adenosylmethionine; DNMT, DNA methyl transferase; NOX, nicotinamide adenine dinucleotide phosphate (NADPH) oxidase; iNOS, inducible nitric oxide synthase; SDM, superoxide dismutase; O_2_^−^, superoxide; H_2_O_2_, hydrogen peroxide; ER, endoplasmatic reticulum.

## Data Availability

Data were found in Pubmed.

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
