# Peer review of "The Controversial Role of HCY and Vitamin B Deficiency in Cardiovascular Diseases"

_nutrients, 2022, doi:10.3390/nu14071412_

Round 1

Reviewer 1 Report

I thank for giving me the opportunity to review the manuscript entitled ‘The controversial role of HCY and vitamin B deficiency in cardiovascular diseases’ submitted to Nutrients.

The topic of the cardiovascular ‘toxicity’ related to hyperhomocysteinemia (HHCy) state was a hot one 2-3 decades ago, unfortunately without real impact on clinical practice at that time, but it is important to update it nowadays in view of the better understanding of the vitamins B deficiency, notably the vitamin B12, as the authors did. This manuscript was globally well-written and pleasant to read.

I just have few issues about this manuscript:

General Comment

The manuscript focused the vitamins B deficiency on folates (B9), vitamin B12, and vitamin B6 but I did not find a word on vitamin B2 (riboflavin) which takes part into the folate-dependent remethylation cycle through the MTHFR while the authors dedicated a long part to MTHFR. I think the manuscript will gain in completeness by dedicating few lines to riboflavin (DOI: 10.1186/s12916-020-01780-x).

Comment on part 1

The authors developed a part to vitamin B12 deficiency in the end of this section. At my end, I think that it would be interesting to do the same for all the B-vitamins related to HHCy.

Comments on part 2

Line 79: the authors cite a meta-analysis from Peng HY et al. reporting that HHCy was not associated with all-cause in men but that it was in women. By looking in details this meta-analysis, we can discover that only two out of the six included studies were used for sex comparisons in the meta-analysis and that the results for men were very different between those two studies compared to those for women, explaining the large confidence interval for men, and so the fact that this is not statistically significant. A more distant reading of these results could state that: 1) they are insufficient to conclude but also 2) that finally the relative risk appeared comparable between men (1.87) and women (1.74). 

Line 102: the authors state ‘The main causes of B12 deficiency are a reduced dietary intake and malabsorption (9)’. Firstly, the topic of the reference 9 is to compare the diagnostic performances of some biomarkers for vitamin B12 deficiency so it appears to me as an inappropriate citation. Secondly, this sentence gives to the reader the impression that this condition of reduced intake is a frequent inducer of B12 deficiency. The main cause of B12 deficiency is malabsorption whatever its cause but isolated reduced dietary intake just represents few percents among the whole causes and it essentially concerns non-supplemented vegetarian/vegan people (DOI: 10.1503/cmaj.1031155).

Lines 109-125: The section about the link between the high vitamin B12 and the mortality is either out the topic of this manuscript or insufficiently developed: at my end, I still think that this part is interesting but the authors shouldn’t report the association between high B12 and mortality without highlighting the association bias related to blood malignancies and solid cancers (DOI: 10.3390/jcm9020474). Without this explanation, this part is confusing for the readers not used to this theme.

Comments on part 3

I will do here a general comment based on my clinical experience: the authors report the association between the polymorphism C677T of MTHFR and the HHCy. My experience is that this polymorphism is not by itself a provider of HHCy and is always associated with a primary cause increasing the level of homocysteinemia, but I agree with the fact that having this polymorphism increases in higher proportion the level of HHCy. The section about the polymorphism of MTHFR in the manuscript lets thinking that the link is direct.

Comments on part 4

General comment: the presentation of the Figure 2 needs to be optimized: the title ‘Figure 2’ appears in the middle of the figure itself and the writing within the blue ellipses is not centered.

Lines 293-294: the authors state that ‘there is also evidence that Hcy-induced ER stress and oxidative stress reinforce each other’ without any reference. A reference should be provided.

Comments on part 6

The authors state, in order to discuss the fact that folates supplementation could or not prevent cardio-vascular diseases, the following sentence: ‘the antifolate Methotrexate has been suggested to reduce CVD risk in rheumatoid arthritis patients beyond the effect of other anti-inflammatory agents’. However, this statement is wrong for two reasons: 1) the paper from van Halm VP reported OR under 1 for each of the immunosuppressive drugs in this study, the fact the confidence interval crosses or not 1 is more about the number of subjects for some conditions (underpowered study for some points?) 2) each patient receiving Methotrexate for immune diseases should also be supplemented with folates (another day in the week that the one they receive the MTX) and my experience in evaluating folates status in all my patients receiving MTX is that deficiency is very rare.

Author Response

  • The topic of the cardiovascular ‘toxicity’ related to hyperhomocysteinemia (HHCy) state was a hot one 2-3 decades ago, unfortunately without real impact on clinical practice at that time, but it is important to update it nowadays in view of the better understanding of the vitamins B deficiency, notably the vitamin B12, as the authors did. This manuscript was globally well-written and pleasant to read.

    We thank the reviewer for this positive overall judgement. Also, in our opinion it is worthwhile updating the scientific community with recent advances in the understanding on the role of HHcy and B-vitamin deficiency in CVD.

  • The manuscript focused the vitamins B deficiency on folates (B9), vitamin B12, and vitamin B6 but I did not find a word on vitamin B2 (riboflavin) which takes part into the folate-dependent remethylation cycle through the MTHFR while the authors dedicated a long part to MTHFR. I think the manuscript will gain in completeness by dedicating few lines to riboflavin (DOI: 10.1186/s12916-020-01780-x).

    As suggested by the reviewer, we have added a comment on the role of B2 in HCY-metabolism and its role in individuals with the MTHFR TT genotype (lines 204-206. In addition, we have quoted the publication mentioned by the reviewer.

    ‘Riboflavin (vitamin B2) is a co-factor of the MTHFR enzyme. B2 deficiency accentuates the loss of enzymatic activity and has been found to increase the genetic risk of hypertension in individuals with the TT genotype.’

  • The authors developed a part to vitamin B12 deficiency in the end of this section. At my end, I think that it would be interesting to do the same for all the B-vitamins related to HHCy.

    In the revised manuscript we have added two paragraphs that address some general aspects of the other B-vitamins that are related to HHcy (lines 59-67 and 77-84)

    ‘5-MTHF is the most abundant folate species in human serum accounting for 82–93 % of the total folate (Pfeiffer CM et al. Clin Chem 50:423–432; Kirsch SH et al. J Chromatogr B Analyt Technol Biomed Life Sci 878:68–75). It provides a methyl group for the remethylation of Hcy to form methionine and tetrahydrofolate (THF). The enzyme 5,10-methylenetetrahydrofolate reductase (MTHFR) converts 5,10-methylenetetrahydrofolate (5,10-MTHF) into 5-MTHF and thus is crucial for the availability of 5-MTHF. The common MTHFR C677T polymorphism is associated with a reduced enzyme activity, which can limit the availability of 5-MTHF in situations when folate supply is low (Smulders YM et al. J Nutr Biochem 18:693–699). Recent evidence suggests that riboflavin in the form flavin adenine dinucleotide (FAD) is a cofactor for MTHFR and may thus represent a key factor linking the MTHFR C677T polymorphism to CVD (Ward M et al. BMC Med. 2020; 18: 318.)’

‘Upon adequate availability of methionine, Hcy is eliminated via the B6-dependent transsulfuration pathway. The term vitamin B6 actually represents a group of six water-soluble compounds with a central pyridine ring, which are involved as co-factors in numerous reactions, mainly in amino acid metabolism and fatty acid biosynthesis (Ueland PM et al. Annu Rev Nutr. 2015;35: 33–70.). Pyridoxal 5'-phosphate (PLP) is the primary metabolically active form. B6 is found in a wide variety of food items including meat, fish and fowl so that deficiencies are rare (Mayengbam S et al. Biomedicines 2020;8(11):469; https://www.nutritionadvance.com/foods-high-in-vitamin-b6/). In addition, intestinal bacteria synthesize B6, which is mainly consumed by non-B6-synthesizing bacteria.’

  • Line 79: the authors cite a meta-analysis from Peng HY et al. reporting that HHCy was not associated with all-cause mortality in men but that it was in women. By looking in details this meta-analysis, we can discover that only two out of the six included studies were used for sex comparisons in the meta-analysis and that the results for men were very different between those two studies compared to those for women, explaining the large confidence interval for men, and so the fact that this is not statistically significant. A more distant reading of these results could state that: 1) they are insufficient to conclude but also 2) that finally the relative risk appeared comparable between men (1.87) and women (1.74). 

    We agree with the reviewer that existing studies are insufficient to draw definitive conclusions on gender differences regarding HCY and all-cause mortality. In the revised manuscript we have addressed this aspect as follows (lines 108-112):

    ‘However, caution is warranted when interpreting potential differences between men and women in this meta-analysis. Only two out of the six included studies were used for sex comparisons with very different results for men, but not women. Therefore, existing studies are insufficient to draw firm conclusions on gender differences.’

  • Line 102: the authors state ‘The main causes of B12 deficiency are a reduced dietary intake and malabsorption (9)’. Firstly, the topic of the reference 9 is to compare the diagnostic performances of some biomarkers for vitamin B12 deficiency so it appears to me as an inappropriate citation. Secondly, this sentence gives to the reader the impression that this condition of reduced intake is a frequent inducer of B12 deficiency. The main cause of B12 deficiency is malabsorption whatever its cause but isolated reduced dietary intake just represents few percents among the whole causes and it essentially concerns non-supplemented vegetarian/vegan people (DOI: 10.1503/cmaj.1031155).

    We thank the reviewer for pointing out this aspect. In the revised manuscript we have corrected this sentence and added some additional information from the paper by Andrès E et al. CMAJ. 2004; 171(3): 251–259 (lines 132-144):

    ‘The main causes of B12 deficiency in elderly individuals are food-cobalamin malabsorption (60%–70%) and pernicious anemia (15%–20%) (Andrès E et al. CMAJ. 2004; 171(3): 251–259). Food-cobalamin malabsorption syndrome refers to the inability to release B12 from food or from intestinal transport proteins in the presence of hypochlorhydria and is frequently the result of gastric atrophy (Carmel R. Baillieres Clin Haematol. 1995; 8(3):639-55; Andrès E et al. CMAJ. 2004; 171(3): 251–259). A reduced dietary intake rarely leads to B12 deficiency (<5%) and is mainly a concern in vegetarian and vegan individuals (Herrmann W, Obeid R. Vitamins in the prevention of human diseases, De Gruyter2011). Cobalamin deficiency is a proceeding process which develops through different stages (Herrmann W, Obeid R. Vitamins in the prevention of human diseases, De Gruyter2011). In the early stages, plasma and cells become depleted of the vitamin causing lowered serum concentration of holotranscobalamin (holoTC) but Hcy and MMA are still within normal range. If the negative balance continuous, the metabolic markers Hcy and MMA become elevated in serum. Pernicous anemia characterized by antibodies against intrinsic factor is the most famous disorder of cobalamin absorption and very frequent among elderly patients (about 50% of the cases).’

  • Lines 109-125:The section about the link between the high vitamin B12 and the mortality is either out the topic of this manuscript or insufficiently developed: at my end, I still think that this part is interesting but the authors shouldn’t report the association between high B12 and mortality without highlighting the association bias related to blood malignancies and solid cancers (DOI: 10.3390/jcm9020474). Without this explanation, this part is confusing for the readers not used to this theme.

    We thank the reviewer for highlighting this argument. In the revised manuscript this has been addressed at lines 167-172.

    ‘In this context, it needs to be mentioned that elevated B12 plasma concentrations are sig-nificantly associated with hematologic malignancies and solid cancers. A large comparison study of individuals with total plasma B12 concentrations above and below 1000 pg/ml demonstrated a strong association of high B12 with solid organ cancer, which was strongest in the presence of metastases (Urbansky G et al. J Clin Med. 2020;9(2):474).’

  • I will do here a general comment based on my clinical experience: the authors report the association between the polymorphism C677T of MTHFR and the HHCy. My experience is that this polymorphism is not by itself a provider of HHCy and is always associated with a primary cause increasing the level of homocysteinemia, but I agree with the fact that having this polymorphism increases in higher proportion the level of HHCy. The section about the polymorphism of MTHFR in the manuscript lets thinking that the link is direct.

    We agree with the reviewer’s view that MTHFR C677T gene mutation is not necessarily a cause of HHcy. In the revised manuscript this has been addressed at lines 197-203.

According to our current understanding MTHFR C677T gene mutation only becomes relevant when either folate or riboflavin status is low. In folate deficient individuals with the MTHFR C677T polymorphism, residual enzyme activity is insufficient to provide enough methyl groups for the remethylation of HCY. Even when folate availability is sufficient, B2 deficiency can further reduce the residual MTHFR activity and thus limit the production of 5-MTHF. However, remethylation of HCY is only of importance when methionine uptake via the food is low.

  • The presentation of the Figure 2 needs to be optimized: the title ‘Figure 2’ appears in the middle of the figure itself and the writing within the blue ellipses is not centered.

    We thank the review for pointing out these technical errors, which have been corrected.

  • Lines 293-294: the authors state that ‘there is also evidence that Hcy-induced ER stress and oxidative stress reinforce each other’ without any reference. A reference should be provided.

    The interaction between HHcy, ER-stress and oxidative stress has recently been reviewed by Fu Y et al. (Br J Pharmacol. 2018;175(8):1173-1189). In the revised manuscript we have referenced our statement with this publication.

  • The authors state, in order to discuss the fact that folates supplementation could or not prevent cardiovascular diseases, the following sentence: ‘the antifolate Methotrexate has been suggested to reduce CVD risk in rheumatoid arthritis patients beyond the effect of other anti-inflammatory agents’. However, this statement is wrong for two reasons: 1) the paper from van Halm VP reported OR under 1 for each of the immunosuppressive drugs in this study, the fact the confidence interval crosses or not 1 is more about the number of subjects for some conditions (underpowered study for some points?) 2) each patient receiving Methotrexate for immune diseases should also be supplemented with folates (another day in the week that the one they receive the MTX) and my experience in evaluating folates status in all my patients receiving MTX is that deficiency is very rare.

    We agree with the reviewer that the MTX argument is rather week. However, it helps to illustrate the controversial discussion on the role of folate in CVD. We also agree with the reviewer that van Halm VP et al. reported ORs and that this study is eventually underpowered. However, the results show substantially lower ORs for MTX than for the other disease-modifying drugs. Of course, the authors did not test if these differences in ORs were statistically significant. We also acknowledge that RA patients treated with MTX are supplemented with folate and deficiencies are rare. However, it remains open to what extend MTX impedes the biological activity of supplemented folate. Therefore, we believe that our statement is still valid and decided to maintain the original wording. In addition, we stated that the results ‘suggest’, but not ‘demonstrate’ an association between MTX and CVD. The studies discussed in the following sentences were rather negative with.

Reviewer 2 Report

The review entitled "The controversial role of HCY and vitamin B deficiency in cardiovascular diseases" is a work that collects the current state of knowledge on the subject of CVD pathogenesis with the participation of homocysteine. The authors cite and analyze a large amount of research while discussing the possibility of clinical / practical application. In my opinion, too little attention has been paid to the role of homocysteine thiolactone, an important intermediate that is considered a physiological substrate for the antioxidant activity of the PON1 enzyme. Points 6 and 7 are a great advantage of the work, in which the authors attempt to systematize the conclusions resulting from the analyzes presented earlier. The downside of the work, in my opinion, is the irresistible impression that its fragments have already been presented in the authors' earlier works. Despite numerous references, I believe that not all parts of the manuscript have been correctly cited.

Author Response

  • The review entitled "The controversial role of HCY and vitamin B deficiency in cardiovascular diseases" is a work that collects the current state of knowledge on the subject of CVD pathogenesis with the participation of homocysteine. The authors cite and analyze a large amount of research while discussing the possibility of clinical / practical application. In my opinion, too little attention has been paid to the role of homocysteine thiolactone, an important intermediate that is considered a physiological substrate for the antioxidant activity of the PON1 enzyme.

    We thank the authors for highlighting this point. In the revised manuscript we have added a short paragraph on Hcy thiolactone and PON (lines 393-402).

    ‘Also, paraoxonase 1 (PON1), a calcium-dependent multifunctional enzyme contained in high-density lipoprotein (HDL) particles, seems to mitigate the pathological effects of Hcy and in particular of Hcy thiolactone (Perla-Kajàn J et al. 2012;43(4):1405-17; Moya C et al. Naunyn Schmiedebergs Arch Pharmacol. 2018;391(4):349-359; Jakubowski H. Adv Exp Med Biol. 2010;660:113-27). Hcy thiolactone is a naturally occurring, particularly atherogenic form of Hcy that arises when Hcy is mistakenly selected by methionyl-tRNA synthetase during protein synthesis. PON1 hydrolyzes aryl esters, lactones, and organophosphates. Due to this function, PON1 detoxifies Hcy thiolactone and thus protects against N-homocysteinylation of proteins. Ultimately, PON1 is an important mediator of the antioxidant properties of HDL, which protect low-density lipoprotein (HDL) from oxidation and thus limits vascular subendothelial damage.’

  • Points 6 and 7 are a great advantage of the work, in which the authors attempt to systematize the conclusions resulting from the analyzes presented earlier.

    We thank the reviewer for appreciating our work.

  • The downside of the work, in my opinion, is the irresistible impression that its fragments have already been presented in the authors' earlier works. Despite numerous references, I believe that not all parts of the manuscript have been correctly cited.

    We agree with the reviewer that this review article also summarizes original research work from our group. In particular, the section on HHcy, B-vitamins and telomeres refers to own results. However, as far as we can judge this is not a copy or anything similar. Should the reviewer refer to other section, please indicate them to us so that we can check.

    Based on the reviewer’s criticism we have checked carefully all references and found several errors. In the revised version of the manuscript, these errors have been corrected.

Reviewer 3 Report

This is a very well written comprehensive review article on the mechanistic actions and efficacy of certain B-vitamins with respect to lowering homocysteine for the prevention of cardiovascular disease.

My only issues with the discussion are on Telomerase. I understand that the steady shortening of telomeres with each replication in somatic cells may have a role in senescence and therefore, many aging-related diseases are linked to shortened telomeres. Organs deteriorate as more and more of their cells die off or enter cellular senescence. However, telomerase is active only in germ cells, some types of stem cells such as embryonic stem cells, and certain white blood cells.  How can you reconcile this with your discussion with respect to CVD?

I also have some formatting/corrections to note:

Page 4 line 127: “CVD” should be CHD.

Page 6 Figure 2: There are a couple of formatting problems that need to be addressed, such as missing arrow heads.

Page 7 lines 300 to 303: “DNA hypomethylation” tends to result in reduced gene expression, but your term “modifies the expression…” needs to be better defined, as well as the phrase “altered expression of telomerase and p21….”.  Which direction were these 2 altered, increased or decreased gene expression?

Page 9 lines 389 to 390: Can you add directionality to the line “where TERT mRNA expression and methylation of the TERT promoter region differed between hyperhomocysteinemic animals and controls.”

Page 10 line 454: 20 micromolar and not nanomolar (20 nmol/L).

Page 11 line 509: Add MRHFR 677TT

Author Response

  • This is a very well written comprehensive review article on the mechanistic actions and efficacy of certain B-vitamins with respect to lowering homocysteine for the prevention of cardiovascular disease.

    My only issues with the discussion are on Telomerase. I understand that the steady shortening of telomeres with each replication in somatic cells may have a role in senescence and therefore, many aging-related diseases are linked to shortened telomeres. Organs deteriorate as more and more of their cells die off or enter cellular senescence. However, telomerase is active only in germ cells, some types of stem cells such as embryonic stem cells, and certain white blood cells.  How can you reconcile this with your discussion with respect to CVD?

    We thank the reviewer for raising this crucial point. The role of telomeres and telomerase in CVD has been reviewed recently by Boniewska-Bernacka E et al. (Exp Cell Res. 2020;397(2):112361). This review is already quoted in the paper. Furthermore, in several experimental studies, Werner C et al. have demonstrated that telomerase is expressed in myocardial and aortic tissue (Werner C et al. Atherosclerosis. 2011;216(1):23-34; J Am Coll Cardiol. 2008;52(6):470-82.). Apparently, telomerase responds to specific stimuli, such as pioglitazone or endurance exercise. However, this aspect is not widely appreciated.

Page 4 line 127: “CVD” should be CHD.

Done

  • Page 6 Figure 2: There are a couple of formatting problems that need to be addressed, such as missing arrow heads.

    In the revised version of the manuscript, all arrowheads are visible.

  • Page 7 lines 300 to 303: “DNA hypomethylation” tends to result in reduced gene expression, but your term “modifies the expression…” needs to be better defined, as well as the phrase “altered expression of telomerase and p21….”.  Which direction were these 2 altered, increased or decreased gene expression?

    As suggested by the reviewer, in the revised manuscript we state that HCY increases p21 expression. Furthermore, we have noted that the wrong reference was quoted. In the revised manuscript we have corrected this: Zhang D et al. Arteriosclerosis, Thrombosis, and Vascular Biology. 2015;35:71–78. See lines 356-359.

  • Page 9 lines 389 to 390: Can you add directionality to the line “where TERT mRNA expression and methylation of the TERT promoter region differed between hyperhomocysteinemic animals and controls.”

    In this study, LTL, TERT mRNA expression and TERT promoter methylation were all lower in the hyperhomocysteinaemic animals. As suggested by the reviewer, this has been specified in the revised manuscript (lines 456-458). The reference quoted in the first version of the manuscript was incorrect and has been updated.

    ‘A causal link between Hcy and telomeres has also been demonstrated in animals, where LTL, TERT mRNA expression and methylation of the TERT promotor region were lower in hyperhomocysteinemic animals than in controls (Zhang D et al. Circ J. 2014;78(8):1915-23.).’

  • Page 10 line 454: 20 micromolar and not nanomolar (20 nmol/L).

    Corrected

  • Page 12 line 574: Add MTHFR 677TT.

Round 2

Reviewer 2 Report

I would like to thank the authors for introducing the corrections, and I also accept the present form of the manuscript.